# The Color of Health: Residential Segregation, Light Rail Transit Developments, and Gentrification in the United States

**DOI:** 10.3390/ijerph16193683

**Published:** 2019-09-30

**Authors:** Shadi O. Tehrani, Shuling J. Wu, Jennifer D. Roberts

**Affiliations:** 1School of Architecture and Environmental Design, Iran University of Science and Technology, Tehran 16846-13114, Iran; niusha.omidvar@yahoo.com; 2Department of Kinesiology, School of Public Health, University of Maryland, College Park, MD 20742, USA; shjwu18@terpmail.umd.edu

**Keywords:** residential segregation, transit-oriented development, transit-induced gentrification, social determinants of health, physical and mental health outcomes

## Abstract

As the modern urban–suburban context becomes increasingly problematic with traffic congestion, air pollution, and increased cost of living, city planners are turning their attention to transit-oriented development as a strategy to promote healthy communities. Transit-oriented developments bring valuable resources and improvements in infrastructure, but they also may be reinforcing decades-old processes of residential segregation, gentrification, and displacement of low-income residents and communities of color. Careful consideration of zoning, neighborhood design, and affordability is vital to mitigating the impacts of transit-induced gentrification, a socioeconomic by-product of transit-oriented development whereby the provision of transit service “upscales” nearby neighborhood(s) and displaces existing community members with more affluent and often White residents. To date, the available research and, thus, overall understanding of transit-induced gentrification and the related social determinants of health are limited and mixed. In this review, an overview of racial residential segregation, light rail transit developments, and gentrification in the United States has been provided. Implications for future transit-oriented developments are also presented along with a discussion of possible solutions.

## 1. Introduction

### “Zip Code Is a Better Predictor of Health”

“Your zip code is a better predictor of your health than your genetic code”, an assertion that acknowledges the overwhelming variance of health and life expectancy among individuals and communities in specific geographical neighborhoods due to socioeconomics, race, ethnicity, and other social determinants of health [1]. In the United States, access and availability of goods and services fluctuate widely along racial, ethnic, and income lines. Historically and contemporaneously, the political allocation of public resources and transportation infrastructure has been significantly associated with residential living patterns. The availability of and access to desired housing and quality of life is governed by specific social policies irrespective of individual actions. Public investment into a community, often well intentioned, tends to increase cost of living and often fails to incorporate existing local culture and necessities. Those with lower incomes or “less compatible cultures” are often forced to relocate to less desirable neighborhoods that lack essential infrastructural elements, such as paved roads, quality schools, safe streets, transportation, and clean water. Transit-induced gentrification (TIG), as a result of transit-oriented development (TOD), also referred to as transit-supportive development or transit-friendly development, is an example of this residential segregating process [2]. As such, these conditions of deprivation are recognized social determinants of health that have had significant impacts on the well-being and quality of life of many communities of color, particularly African Americans.

## 2. US Racial Residential Segregation

### 2.1. Suburbanization in the 20th Century

In the late 19th century, racial segregation was by census block; however, by the middle of the 20th century, this type of residential segregation changed to the neighborhood level [3]. Within and by each neighborhood, White and affluent families were separated from poor African American residents [4,5]. After World War II, a population expansion instigated a suburban boom in nearly every metropolitan area of the United States. City overcrowding and slum conditions, along with the development of suburban service delivery, commuting possibilities, and rising incomes motivated wealthy White families to move out of the congested city center [6,7]. While suburban development across the United States was directly related to the evolution of transportation routes, suburbanization was influenced by technological developments and social ideologies of racial residential segregation, which stemmed from Black Codes and Jim Crow laws. Black Codes were laws designed to limit the freedom of African Americans and ensure their availability as a cheap labor force after the emancipation of slavery. Additionally, Jim Crow laws enforced racial segregation and second-class citizenship for African Americans between the end of reconstruction in 1877 and the beginning of the civil rights movement in the 1950s [8,9]. Further spurring this racial residential segregation, zoning laws and mortgage insurers or guaranteers, including the Federal Housing Administration (FHA) and US Department of Veteran Affairs (VA), denied African American homeownership in most suburban subdivisions [10,11]. Since racially integrated neighborhoods were not considered wise investments by the FHA or VA, African Americans and other individuals of color were generally excluded from participating in this suburban housing boom [12]. Therefore, they were relegated towards older and declining housing stock in the urban centers, which had been abandoned by the White middle class. Compounding the issue of deteriorating homes, residents escaping to the suburbs withdrew their support from public services in the city and further perpetuated these exclusionary zoning practices [13]. Consequences of suburbanization and segregation led to the formation of ghettos and slums, rise in homelessness and neighborhood poverty, and overall economic decline in urban central business districts predominately inhabited by African Americans and other racial/ethnic minorities [14,15,16].

### 2.2. Determinants of Racial Residential Segregation

Between 1900 and 1940, racial residential segregation occurred through more than half of the United States. For example, in 1900, nearly all African Americans lived in a few southern states, which contained only 25% of all White Americans. During this time, the dissimilarity index, a metric for measuring residential segregation between African Americans and Whites, was 64 between states and 69 between counties, thus indicating that approximately 70% of all African Americans would have had to change their county residence in order to achieve a uniform racial distribution across county lines [17]. By 1940, the dissimilarity index between African Americans and Whites dropped to 52 at the state level and 59 at the county level, yet, indices within cities sharply increased (e.g., 86 Cleveland, OH, USA; 82 Buffalo, NY, USA), which suggested the progressive formation of African American ghettos throughout American cities [17]. These dissimilarity indices were arguably a function of the Home Owners’ Loan Corporation (HOLC). Created in 1933 during the Great Depression and under New Deal legislation to help those in danger of home foreclosure, HOLC was a government sponsored corporation responsible for creating “Residential Security Maps”, commonly known as redlining maps, of major American cities [10]. Mandated by the national ethics code of segregation, HOLC considered neighborhood racial composition to gauge mortgage lending risk, whereby the safest neighborhoods were colored green and the riskiest neighborhoods were colored red. If an African American lived in a neighborhood, it earned a red color, hence redlined. Even before the surge of American suburbanization in the 1950s, these maps documented how loan officers and appraisers evaluated mortgage lending decisions through institutionalized racism. Until 1970 when it reached historically low levels, racial residential segregation at the state and county macro-levels demonstrated a downward trend [17]. However, at the neighborhood level, racial residential segregation showed the opposite pattern, albeit in a decelerating pace [17]. While the trend for race residential segregation decreased, class residential segregation remained static [18,19].

Several explanations for the depth and persistence of residential segregation have been discussed from different perspectives. Some studies highlighted the role of income and class, same race preference among African Americans, education, and socioeconomic status on the perpetuation of racial segregation [17,20,21]. Yet, other research debunked these explanations and proved them to be unsubstantiated and inconsistent [5,20,22,23]. In a recent study, racial and geographical segregation was shown to be much more complicated than economic injustice and racial antipathy and is heavily influenced by local politics [24]. Since the beginning of American history, most investors and homeowners have been White individuals, who, as mentioned previously, perceived the presence of African Americans and any individuals of color in their area to be a negative reflection of their home’s value. Along these lines, landowners prefer homogenous White neighborhoods because they were more likely to reinforce market economy [25]. Local governments also supported the private sector by dictating biased public policies in favor of land-oriented businesses to benefit from higher taxes. The quality of services and amenities, such as parks and playgrounds, access to healthcare, and developed infrastructure, all depend on the decisions of political stakeholders. Thus, local government initiatives are empowered to increase the economic opportunities and property values of White homeowners, which ultimately preserves a culture of White homogeneity, socioeconomic inequalities, and residential segregation along the race and class lines.

### 2.3. Urban Planning and Residential Zoning

The distribution of public goods at the city level is one of the main factors behind the rise in property and housing prices, which incentivizes zoning practices for government and the private sector. Urban planners initially proposed zoning as a remedy to the problems triggered by industrialization. Some of the main zoning objectives are to promote the health, safety, and quality of life for communities; encourage the most appropriate use of the land; and regulate residential density. However, exclusionary zoning ordinances contributed to the historical and contemporary patterns of profound racial segregation that characterize American neighborhoods today. Exclusionary zoning has been defined as “local land-use controls that have the effect of excluding most low-income and many moderate-income households from suburban communities and, indirectly, excluding most members of undesirable populations (e.g., people of color, poor people, immigrants) [26,27]. In practice, zoning tended to preserve the status quo through exclusionary zoning and deed restrictions, or restrictive covenants, both acting to eternize Jim Crow segregation [28]. Discriminatory planning and exclusionary zoning contributed to unequal development within metropolitan areas, therefore limiting access of all citizens to affordable housing, public transportation, thriving school systems, and economic infrastructure. As a result, the formation of segregated communities along race and class lines and the creation of an urban underclass that was denied access to mainstream opportunities were reinforced [5,29,30].

Zoning is employed continuously in the United States, and through these practices, the pattern of political geography exacerbates social inequities and ultimately limits the spatial distribution and accessibility of quality public goods and services. For example, transit deserts, a concept similar to food deserts, occur when the demand for public transportation exceeds supply. Due to the inequitable spatial distribution of transportation services, some regions may be well endowed with transit options while others may not. These transit deserts are generally characterized by poor public transit planning, a function of inadequate TOD and poor zoning (e.g., mass transit dead zones). TODs, which are designed in varying shapes, sizes, formations, and environmental compositions, possess the common element of transportation interface. As described later in Section 3.2, a sophisticated typology of fixed transit precincts, born from the conceptualization of TOD, has been developed. However, for the purpose of this review “TOD” will be used, unless otherwise noted, as either a conceptual term referencing its early definition—community development centered on transit facilities—or as a defined transit precinct with high density and grid-street patterned residential and commercial centers designed to maximize walking, biking, and public transit [31]. TOD, along with zoning policies not only influence housing and rental values or public transit options, but both can also instigate changes in neighborhood demographic patterns, the share of local service land, and the appearance of a community.

Measures restricting African Americans and other communities of color often assure that neighborhoods in which they live remain disinvested and disadvantaged [5,10]. The allocation of funds to specific areas and projects creates racial residential stratification, which can either support or obstruct social determinants of health depending on racial stratum. According to Health People 2020, “social determinants of health are conditions in the environments in which people are born, live, learn, work, play, worship, and age that affect a wide range of health, functioning, and quality-of-life outcomes and risks” [32]. While the contextual disadvantages in these disinvested neighborhoods provide an excellent opportunity for public investments and renovation, such as TOD, there are often unintended consequences that aggravate and widen the gap to already existing inequities.

## 3. US Light Rail Transit Developments and Gentrification

### 3.1. Historical Snapshot of Public and Private Transportation

The streetcar was one of the most important transit-accessible services in the United States from the late 19th until the early 20th centuries. Streetcar lines developed American cities by defining residential and commercial areas. The tax revenue generated from rising land prices around streetcar tracks was spent on improving urban infrastructure [33]. By the mid-1960s, with the construction of highways, the allure of suburbs, and the demand for faster and more flexible transportation, the streetcar began to disappear from cities in the northern United States. The early beginnings of urban sprawl became apparent with the construction of new roads and freeways at more distant destinations and patterns of low-density development. As affluent White residents became more private-car-dependent, low-income and people of color suffered from fewer mobility options [34,35]. In the early 1980s, intensified urban sprawl, suburbanization, and over reliance on automobiles, along with the accompanying issues of traffic congestion and air pollution, motivated urban designers to launch the TOD movement. Scholars found rail-based transit as one of the main ways to cope with auto-dependent urban planning. As follows, the electric streetcar of pre-World War II was re-introduced in the shape of light rail transit (LRT) [36].

### 3.2. Land Value Uplift and Light Rail Transit

For many reasons (e.g., mass transit expansion, urbanization), public transit use has increased among Americans and tends to be higher among African Americans [37,38]. The greatest increase occurred with LRT, a 280% increase in passenger miles from 1990 to 2010 [39,40]. Characterizations of LRT include electric trains running along fixed routes with dedicated track corridors and passenger boarding stations [41]. With smaller cars than commuter trains and traffic signal priority to ease efficiency, LRT has greater utility for implementation in dense metropolitan areas [42,43]. Used mainly as a way to promote accessibility of low- and moderate-income residents to employment, retail, recreational activities, and motivating revitalization in neglected areas, LRT development has been particularly popular in recent decades [44]. Recognizing the potential for rapid connectivity, 30 American cities have planned, built or expanded roughly 50 LRT systems [45,46].

The LRT station areas, serving as TODs, are strategic targets for developers to build condominiums and upscale housing, which is likely to increase property values and rents in the neighborhoods [47]. Emerging as a popular and influential planning concept and coined as a concept by Peter Calthrope [31], TODs include a mix of commercial, residential, and entertainment properties centered around or located near a transit station [48]. Beyond this conceptual definition, TODs are characterized by pedestrian-focused built environments with grid street patterns; high population density; designated parking; bicycle access and storage; and multi-family homes and office and retail spaces that are vertically and horizontally oriented. Conversely, transit-adjacent developments (TADs) or the TOD-TAD hybrids are much more auto-dominated, industrial and/or segregated with respect to land-use area. For example, when compared to TODs, TADs have been found to be nearly four times further away from central business districts and considerably less walkable due to the block length and intersection density. While TODs, TADs, and TOD-TAD hybrids fall within a built environment spectrum, all of these fixed transit precincts encompass varying social environment features as well. Research has found that approximately 75% of TOD households are renters as compared to less than 50% of TAD households [48]. Furthermore, TOD households had a median income of $17,000 less than TAD households based on available 2010 data [48]. Among the 4399 transit stations in the United States, 31.8%, 37.3%, and 30.9% were categorized as TADs, TODs, and TOD-TAD hybrids, respectively [48]. As previously defined, both built and social environment features describe a typology of all fixed transit precincts.

A number of empirical studies in the United States have shown that the provision of new LRT has a significant impact on increasing neighborhood housing values, population densities, and fixed capital investments [49,50,51]. Using multiple regression, associations between transit accessibility and house prices were found, independent of other house or neighborhood features [41,51,52,53]. Hess (2007) [47] identified a positive impact of LRT stations on residential property value in Buffalo, NY, USA. Specifically, a home located within a one-quarter mile radius of an LRT station earned a premium of $1300–$3000 USD, or 2–5% of the city’s median home value, even though effects were not found evenly throughout the entire LRT system of 14 stations. A study in Saint Louis, MO, USA examined the impacts of a LRT line on residential property values. The results showed that proximity increased value at $14 USD per foot closer to the LRT station, for properties within a quarter of a mile [47]. In Portland, OR, USA researchers found that single-family homes near an LRT station gained much higher premiums, and property values declined by approximately $30 USD for each meter away from stations, beginning at a distance of 100 m [54]. Similarly, the prediction of the future gain from an LRT line before construction has demonstrated a positive impact on house prices in other studies [53,55]. Price premiums have also been reported in higher-income neighborhoods and those far from the central business district served by rail transit. For example, the housing premium associated with TOD in San Diego, CA, USA, which included TOD, TAD, and TOD-TAD Hybrid stations, reported that a condo in a pedestrian-oriented environment and near a LRT station had a significantly higher value than a condo in a similar neighborhood and not near a LRT station [56]. While most studies have identified positive associations between property values and LRT, the research findings have not always been consistent. In Atlanta, GA, USA, houses adjacent to LRT stations have been found to have lower values compared to those which are farther, due to the increased traffic, noise, and crime rate [52]. A negative effect, by way of increasing rent and declining property value, was also reported in San Jose, CA, USA [57]. The declined value was nearly $2 USD per meter of distance between a house and the nearest LRT station [57]. However, transit network advancement, regional growth, and increased road congestion incited a later increase in house price in Santa Clara County, which includes San Jose, CA, USA [51].

### 3.3. Defining Gentrification and Transit-Induced Gentrification

The term “gentrification” was first coined in 1964 by British sociologist Ruth Glass as a way to describe “one by one, many of the working class quarters of London hav(ing) been invaded by the middle classes-upper and lower…. once this process of “gentrification” starts in a district it goes on rapidly until all or most of the original working class occupiers are displaced and the social character of the district is changed” [58]. Glass further made a distinction between gentrification and redevelopment by stating that that unlike gentrification, “the process by which working class residential neighbourhoods are rehabilitated by middle class homebuyers, landlords and professional developers”, redevelopment “involves not rehabilitation of old structures but the construction of new buildings on previously developed land” [58]. Interestingly, this latter portion of the Glass definition conflicts with the present-day images of gentrification. While pinning down a contemporary characterization for gentrification has been challenging over the past several decades, it has commonly defaulted to the Glass definition. Considering that the gentrification process has evolved with 21st century influences, an extended interpretation, which acknowledges a much larger phenomenon, the class remake and new building of urban residential landscapes, is unavoidable. As such, a wider representation that focuses on four core elements of gentrification has been offered [59]. It has been suggested that a contemporary characterization of gentrification, which concentrates on four core elements (i.e., (1) capital reinvestment; (2) social upgrading of locale by incoming high-income groups; (3) landscape change; and (4) direct or indirect (exclusionary) displacement of low-income groups) and disassociation from particular landscapes or contexts can be a significant way to conceptualize and analyze urban change in the 21st century [59]. Even with this augmented annotation, the central focus of gentrification debates pertains to the impacts on new and existing neighborhood residents and the question of who can benefit the most from new developments. A considerable amount of work portrayed gentrification as “a class-based process of capital reinvestment through which middle-class individuals and interests stake claims to urban communities after a period of economic disinvestment and alter the physical and social milieus to suit their preferences” [51]. Cultural preferences, political orientations, and economic development are the basis of gentrification.

The main agents of urban renewal in the United States and elsewhere typically are White, middle-class, urban professionals often working in real estate and government [52,53]. Furthermore, the role of government serves, through housing and infrastructure investment, at the local and state levels, while the private sector tends to manage land-use and city development to accumulate capital [54,55]. Local policies decide where to direct public and private capital for housing, block amenities, and even TODs [60]. Even though public transit is an alternative mode of transportation to promote social equity for low-income people or those unable to afford and maintain personal vehicles, TODs often trigger a “back-to-the-city” wave of high-income households due to the increased land-use intensity and transit accessibility that result from these developments [61,62]. Neighborhoods start to experience disproportionate increases in the number of White, young, well-educated, middle- or high-income professionals and small families [63,64,65]. With the movement of affluent households from the suburbs to city centers and the relocation of disadvantaged groups to suburban areas, broader shifts can happen in the spatial distribution of neighborhood advantages within a metropolitan area. As a result, undervalued areas potentially gain value. This ultimately closes the gap between potential value and where the land prices increase, thereby giving rise to gentrification [66].

Impoverished neighborhoods and communities of color often bear the brunt of unintended TOD impacts [67,68,69]. Several studies have characterized TOD impacts as promoting economic development, elevating property values, and enhancing livable environments [56,61,70,71]. However, these positive benefits, as well as the negative impacts, of TODs are not equally distributed. Transit-induced gentrification (TIG), a socioeconomic by-product of TOD, is defined as a phenomenon whereby the provision of transit service, such as LRT, and associated area of development change in the direction of neighborhood “upscaling” [72]. With rising property values and loss of affordable housing, displacement, segregation, polarization, and social loss have been documented as unfavorable TIG externalities [73,74,75,76]. Furthermore, research has shown that transit station neighborhoods are particularly prone to TIG and displacement of existing residents [61,72,73,74,77,78]. As an example, in 42 neighborhoods within 12 metropolitan areas that were first served by rail transit between 1990 and 2000, negative impacts of TOD, specifically the introduction of LRT stations, were observed through research analysis. While there was no fundamental change in neighborhood racial composition, rapid rises in rent and owner-occupied units were found, which resulted in more expensive housing stock, wealthier residents, and increased vehicular ownership [79]. The (in)equitable transit provision was assessed from different aspects, including race, gender, class, and social disadvantage [80,81].

### 3.4. Research Examining Transit-Induced Gentrification

The process of TIG is mostly understudied and existing studies deliver mixed results. Yet, research in this area is particularly important in order to inform regulating policies to minimize the marginalizing effects of TIG. Of particular difficulty is determining the proper parameters to define and observe the phenomenon of TIG in research. Recent studies have utilized changes in household income, house values, new house construction, and educational levels of the community surrounding a transit center as measures of TIG [73,82,83]. A systematic review of 35 research studies presenting evidence on TIG outcomes resulting from transit-based interventions concluded that proximity to transit may contribute to TIG [84]. Research to date has also evaluated the impact of gentrification on neighborhood displacement, believed by some to be an additional dimension or precondition of gentrification, and found no significantly consistent evidence [64,85,86,87,88]. Yet, these studies have not been without limitations, often due to data constraints that require defining neighborhoods as very large spatial aggregations, defining gentrification too broadly, or examining mobility over relatively long intervals of 10 years [89]. In the previously referenced research, displacement was measured separately, as evidence of gentrification does not necessarily indicate that displacement has occurred [73,82]. Displacement can occur throughout the TIG process and is dependent on the number of owner occupied units, demand for housing, vacant units, and ethnic mix of the neighborhood. Hence, displacement needs to be measured contextually. To illustrate, it was found that despite ongoing TIG, displacement was more likely occurring in racially integrated neighborhoods than neighborhoods composed primarily of White residents.

Some research focused on one aspect of gentrification, such as the change in the racial composition or poverty level in a neighborhood [73,90,91]. For example, the analysis of 14 urbanized areas within the United States measured socioeconomic status change in residents before and after building LRT stations [92]. These urbanized areas were selected because their LRT systems started operations by or before 2000. In Denver, CO, USA and San Francisco, CA, USA the rise in the percentage of White, wealthier, and highly educated residents near LRT stations was a key sign of the gentrification process. The presence of the LRT had a great impact on gentrification in Denver’s station-tract neighborhoods, which experienced a 4% relative increase in the White population compared to non-station tract neighborhoods [92]. Moreover, there was a 26% relative increase in the neighborhood change index in LRT station neighborhoods as compared to non-station area neighborhoods. These combined results could indicate that Denver’s LRT station areas experienced gentrification-related neighborhood change in the absence of sustainable transportation promotion by way of TOD. There was also pronounced gentrification and TOD related changes in San Francisco, CA, USA between 1980 and 2010. Even with the positive TOD effects of retaining commute mode shares by public transit and non-POV (non-privately operated vehicle) mode, station tracts in San Francisco, CA, USA exhibited a significant relative increase in income (+31%), neighborhood change index (+9%), White population (+4%), and a decrease in poverty (−3%) [92]. Conversely, Portland, OR, USA exhibited counter-gentrification with a decrease in neighborhood change index (−28%) and increase in poverty (+4%). It was inferred that the counter-gentrification in Portland’s LRT station areas was due to more residents with high transit needs being able to occupy LRT station areas. In Los Angeles, CA, USA and Buffalo, NY, USA it was also found that lower-income residents were occupying neighborhoods in close proximity to LRT stations and that these census tracts revealed a relatively higher poverty rate than other districts [92]. However, unlike Portland, OR, USA where improved transit access largely benefited the low-income residents, this was not observed in Buffalo, NY, USA or Los Angeles, CA, USA due to the declining station neighborhoods. Other research found that percentages of African Americans and Hispanics reduced in racially integrated neighborhoods where there was TOD [73,91]. Similarly, a recent study of LRT neighborhoods in Seattle, WA, USA demonstrated a change in the ethnic–racial composition of the census tract near train stations in both central-city and suburb context [93]. The opening of a new LRT with different quality within and outside Seattle, WA, USA triggered gentrification. In 1980, the city center provided a more diverse population with about 30% of residents who were people of color and the suburban areas consisted of almost all-White residents. However, throughout 1980 to 2014, suburban LRT-treated neighborhoods saw a 50% decrease in the White share of neighborhood residents. Also, through a reciprocal process, the percentage of the African American population in neighborhoods nearly doubled and both Asian/Pacific Islander and Hispanic residents increased to approximately 19% by 2014 [93]. The analyses demonstrated that affected neighborhoods in Seattle, WA, USA experienced rising shares of White residents following the start of LRT construction, while neighborhoods at the perimeter of the line endured a significant increase in racial and ethnic diversity [93].

In general, some scholars argue that gentrification generates desirable results such as social mixing to increase social capital and cohesion [94,95]. Specifically, an influx of the middle class with different educational levels may add diversity and allow public investment into neighborhood infrastructure. Advocates often regard such investments as a way to upgrade infrastructures and housing values, which can have beneficial outcomes for pre-existing residents. Notable are improvements in crime levels, neighborhood aesthetics, amenities, and resources. Although the extent to which public investments, such as TODS, initiate or accelerate TIG and residential displacement is controversial, it is important to recognize that government regulations also play a role in displacement patterns. Simulation studies in Washington, DC, USA metropolitan neighborhoods demonstrated that displacement is most likely in TOD areas with the least amount of government regulation, regardless of the TOD typology (e.g., TOD, TAD and/or TOD-TAD Hybrid) [82]. Taking into account these variables, it is apparent that TIG research is complex and requires continual careful development. Research currently offers only mixed support for claims that TIG results from TOD [82]. Aptly, debates about and the results of TIG research is multidimensional and developing.

## 4. Health Impacts and Consequences of Gentrification

### 4.1. Health Outcomes and Health Determinants

Transit-induced gentrification can engender health consequences when built and social environments are rapidly transformed [82]. Studies have found that populations displaced by gentrification, as compared to those who remained, typically have shorter life expectancy, higher cancer rates, more birth defects, greater infant mortality, and higher incidence of asthma, diabetes, and cardiovascular disease [75,96,97,98,99,100,101,102,103,104,105,106]. Using publicly available data for New York City, NY, USA an adverse association between gentrification and preterm birth among African American residents was found. Specifically, among African American residents a very high gentrification was adversely associated with preterm birth (AOR = 1.16; 95% CI: 1.01, 1.33) as compared to those who lived in a very low gentrified neighborhood [103]. Yet, living in a very highly gentrified neighborhood was protective as compared to living in a very lowly gentrified neighborhood (AOR = 0.78; 95% CI: 0.64, 0.94) for White residents, thus defining a distinct effect of gentrification by race and ethnicity [103]. Hypertension, one of the strongest risk factors for cardiovascular disease, has also been found to be inversely associated with neighborhood affluence/gentrification (OR = 0.7; 95% CI: 0.6, 0.9) among adults participating in the Chicago Community Adult Health Study [105,107]. Conversely, the risk of displacement was positively associated with hypertension (PR = 1.25; 95% CI: 1.08, 1.46) and hypercholesterolemia, another risk factor for cardiovascular disease, (PR = 1.12; 95% CI: 1.01, 0.24) among a population of Hispanic renters in high foreclosure areas (Chicago, IL, USA; Miami, FL, USA; New York City, NY, USA; San Diego, CA, USA) [108]. Gentrification has also been found to be associated with worse self-rated health in neighborhoods with an increasing number of African Americans [109]. In particular, the authors stated “that while gentrification does have a marginal effect improving self-rated health for neighborhood residents overall, it leads to worse health outcomes for [African Americans]” [109]. Similarly, gentrification was associated with poor self-rated health across African American residents in California. By analyzing data from the California Health Interview Survey, gentrification was significantly associated with fair/poor self-rated health (AOR = 2.44, 95% CI: 1.36, 4.37) among those self-identifying as African American [110]. For White residents in gentrifying neighborhoods, gentrification, albeit insignificant, was associated with improved self-rated health (AOR = 0.73, 95% CI: 0.52, 1.01) [110]. Again, a distinct difference of the impact of gentrification was identified between African American and White residents. It is important to acknowledge that even though some research has identified a positive relationship between gentrification and health, it may be due to the fact that they mainly focused on the remaining residents of the urban renewal and not those harmed from spurred gentrification and displacement [65]. Another explanation for improved self-rated heath in the gentrified neighborhood is that the residents who were sampled were the affluent new residents and not those who had been displaced [111,112]. Mental health outcomes, including an increased risk of psychological stress levels and depression, have also been demonstrated among displaced populations [75,96,98]. The mental health impact related to social loss or the disruption of long-time residential ties and the sense of community diminishment can deteriorate a neighborhood’s resilience by weakening social networks [113,114,115]. In a cross-sectional study examining the impact of residential displacement on mental health within gentrifying and non-gentrifying neighborhoods from 2010 to 2014, displaced residents were more likely to be diagnosed with mental health related conditions (37% versus 18%) before baseline [98]. Another study found that multiple displacements raise the level of psychosocial stress and death rate of obesity-related disorders such as diabetes among African American residents [116].

Health determinants related to gentrification and more specifically TIG include limited availability of affordable housing, increased walkability, and as well as a reduction in crime [96,98]. Among communities of color in particular, health inequities such as exposure to underserved social and physical environments, absence of healthy foods, higher risk of violence and crime, and limited housing choices are the long-term negative health effects of displacement [112,117]. Although the availability of walkable streets during the construction period of TOD may be limited, the use of LRT has been found to be associated with an increased likelihood of walking [118]. In regard to rates of crime and gentrification, this relationship has yielded inconclusive findings over the past several decades. A time-series analysis of crime rates between 1970 and 1984 in 14 gentrified neighborhoods throughout Boston, MA, USA; New York, NY, USA; San Francisco, CA, USA; Seattle, WA, USA; and Washington, DC, USA indicated some eventual reduction in personal crime rates but that there was no significant effect on property crime rates [119]. Furthermore, TIG perception or rather the perception of adverse neighborhood changes among residents has also been found to be associated with many of the aforementioned health outcomes and determinants [120,121]. For example, a cross-sectional analysis found that the perception of neighborhood problems and changes were strongly associated with more smoking and hypertension, two risk factors for cardiovascular disease [121]. Additionally, worsening neighborhood perceptions were associated with small increases in depression in a repeated cross-sectional study [120]. Cross-sectional analyses reported that both men and women who reported positive neighborhood changes in transit convenience were twice as likely to increase their walking [122].

### 4.2. Geographically-Patterned Inequities and Inequalities

Gentrification is an urban political, economic, and social process engaged with racial segregation, inequity and inequality, particularly in the United States. Inequity and inequality, terms, terms which are often used interchangeably, are both appropriate for use when discussing gentrification. “Inequity refers to unfair, avoidable differences arising from poor governance, corruption or cultural exclusion while inequality simply refers to the uneven distribution of health or health resources as a result of genetic or other factors or the lack of resources” [123]. Gentrification can result in inequities due to the “poor governance” over development projects thereby leading to additional inequalities in goods, services, and other resources. As geographically-patterned inequalities and inequities expand in neighborhoods, more African American residents can experience alienation, which affects social networks and collective efficacy, factors related to health and well-being [124,125,126]. A considerable amount of work has highlighted the role of gentrification in racial health disparities [103,109,127]. Yet, a vast majority of the literature gives a complicated story of the relationship between gentrification and health-related outcomes, thus implying that the impact of gentrification can vary among individuals and communities. Direct exposure to racism, microaggressions, and segregation in predominantly White neighborhoods have been shown to increase the stress and reporting of poor health during gentrification among African American residents [128]. Also, the scale in which gentrification occurs is quite important. Revitalization of a few neighborhoods within a city and the shift of low-income residents to another tract cannot improve the city-wide health inequalities. Gentrification catalyzes residential displacement and clusters disadvantages outside of the gentrifying neighborhoods in which incumbent households moved.

The impact of gentrification among communities of color is discussed from three aspects: (1) residential displacement; (2) cultural displacement; or (3) disruption of local community ties [109]. The cost of new neighborhood amenities and rental and housing prices create a constant feeling of intimidation, resulting in psychological distress among incumbent residents [129,130]. Gentrification can also put long-term residents under financial and social pressures due housing vulnerabilities and homelessness, restricted access to food or medical care, or feelings of disenfranchisement [131]. With respect to cultural displacement, new businesses that replace “mom and pop” or small family-owned stores and restaurants may diminish the sense of place and sense of community for long-standing residents. Feeling unwelcome in neighborhoods with race-based changes to local establishments explains the higher stress of incumbent residents [130,132,133]. Research in New York City, NY, USA reported that gentrification was associated with a higher stress levels as a result of cultural and historical asset losses, racial biases of incoming residents, and police intimidation for the African American population [134]. Furthermore, gentrification disrupts social ties among long-standing residents, causing a feeling of isolation and waning social health [133,135].

Health inequities and inequalities from gentrification can vary for different age groups. It has been shown that aging adults prefer to remain in their current home and neighborhood, thus benefiting from place attachment, social ties, and social support, which promotes their health and wellbeing [136,137]. Yet, economically vulnerable older adults are prime targets for landlords who seek more benefit from urban renewal efforts. Consequently, this population of adults has experienced more depressive and anxiety-related symptoms in gentrifying neighborhoods due to the fear of increasing housing costs and displacement [5,117,138]. On the other end of the age spectrum, there have been different pathways by which gentrification impacts the development of children. Gentrification affects children’s school readiness and achievement and dropout rates; emotional difficulties; and adolescent pregnancy [139]. For children, residing in disadvantaged neighborhoods can put them at a higher risk of drug use, violence, crime, and victimization [140,141]. However, in affluent neighborhoods, children of color may be excluded from privately provided services and encounter inequities in access and quality of neighborhood resources [139,142].

## 5. Implications for Future Transit-Oriented Developments

A range of approaches can be used to mitigate the process of displacement in gentrified neighborhoods and ensure that existing residents remain intact. Local government can fund projects and support social equity initiatives to improve the scope of community, social, and economic development. For example, affordable housing strategies play a crucial role in reducing exclusionary displacement. Simulation studies in metropolitan Washington, DC, USA communities conducted by Dawkins and Moeckel (2016) [82] show that imposing region-wide housing restrictions (limited to a 15% increase) was more effective than TOD-specific housing restrictions in preventing elevating housing costs. In addition, this same study found that affordability restrictions placed on TODs worked better than housing vouchers for keeping low-income families closer to transit stations. Furthermore, housing vouchers were unable to meet the demands of elevating housing prices. This study recommends that social policies focus on developing affordability requirements for TODs while simultaneously promoting mixed-income households [82].

Another government strategy is to reserve low-priced land at an early stage of TOD in order to provide the grounds for the construction of affordable housing [143]. Reserving and protecting land before gentrification occurs can assure affordable housing units for low-income households when land and housing prices begin to rise. Careful regulations, such as inclusionary zoning, mixed-rate, or mixed-use housing policies, can encourage for-profit developers to finance affordable and mixed-income housing. Also, property tax relief programs and rent subsidies based on income assessments are two other strategies. Property tax relief programs support owner-occupied units while rent subsidies support renters in managing daily costs [143]. Overall, governmental and non-governmental policies, programs, and financing tools that support the creation of mixed-income communities surrounding transit stations are essential to ensure that the benefit of new transit investments are distributed equitably and efficiently [144].

Local communities, by announcing their involvement in neighborhood projects, can also employ actions to support smart growth development. Community stakeholders and local council members need to actively participate in neighborhood planning and understand their rights in order for appropriate and well-informed negotiation to occur with developers. Using a three-level deliberation framework (i.e., (1) inform; (2) involve; and (3) collaborate), stakeholders can establish inclusive housing strategies for equitable and sustainable TODs [145]. At the first level, planners need to inform stakeholders about existing affordable housing policies and strategies and provide examples for how these initiatives have been developed in other areas. Stakeholders must then be involved in the deliberation of identifying challenges to inclusive TODs at the second level, which is then followed by the third level of collaboration to identify solutions. For example, one solution presents itself in the form of Individual Development Accounts (IDAs) programs. These IDAs, a vehicle for homeownership, can provide education and economic resources to lower-income residents, thus improving their ability to purchase a home in their current neighborhood [146]. Other solutions, such as developing vacant or under-used parcels within existing neighborhood areas through infill development can offer economic opportunities, expand homeownership, mixed-use development, and encourage investment in infrastructure and amenities [147]. To ensure the success of these solutions, regulatory policies should be in place to control the private housing market. Finally, stabilization policies such as zoning, land use, and rent control are a set of measures to reduce the volatility of the market and encourage welfare-enhancing growth [148]. Whether the TOD project and resulting gentrification is defined as a positive or negative phenomenon depends on the actions of all stakeholders. In many cases, negative effects can be mitigated by appropriate and timely decisions. Local government, through the use of precise policies and strategies at an early stage of TOD, as well as monitoring changes and working with planners and developers throughout the process, can help create TODs that promote inclusivity, social justice, diversity and the health and well-being for all.

## 6. Conclusions

Throughout American history, evidence has revealed that local governments have rigidly pursued different policies to ensure long-standing segregation along both race and class lines. By employing zoning practices and strategically allocating public goods, the generational wealth through property ownership was secured for White Americans, which consequently led to increased polarization and segregation. Even today, the decisions that are made for urban renewal are drawn from the goals and interests that had supported residential segregation and institutionalized racism in past decades. Some TOD projects and investments, such as LRT, may inadvertently induce TIG though increased home and rental prices, and displace incumbent residents, thereby reinforcing inequity, inequality, and social divisions experienced in the United States. The benefits of LRT including reduced traffic congestion, affordable public transportation, reduced pollution, increased walkability, and economic development can and should be beneficial and inclusive to all, particularly among the pre-existing residents in which LRT developments occur. Therefore, it is necessary for local government and city planners to adopt policies and for communities to be empowered with tools that could thwart gentrification and the unintended consequences of LRT development.

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
