# Peer review of "The Color of Health: Residential Segregation, Light Rail Transit Developments, and Gentrification in the United States"

_ijerph, 2019, doi:10.3390/ijerph16193683_

Round 1
Reviewer 1 Report
In this review of relevant literature, the authors dive into the often-puzzling relationship among such urban forces as recent public transportation investments in local communities, gentrification of neighborhoods, displacement from gentrifying neighborhoods and the health outcomes of residents of these areas, both those who leave and those who are able to stay behind. This is a thorough and mostly complete literature review, one that should be references and cited by scholars seeking to examine how political and economic forces like transportation policy and infrastructure affect our health and well-being.
In short, transportation policy may be color-blind when it is written on paper, but as the authors show, there is consensus in the field that where people live and what assets exist in their communities exerts a powerful effect on health outcomes. Given how historical zoning and planning policies created intense segregation over the 20th century in American cities, it is crucial to continue updating this field of research to understand how more recent urban development alters (or in some cases, does not alter) the very basic question of where we live, and why.
The authors make an implicit claim that public transportation, such as light rail, can be conceptualized as a social determinant of health, and given the evidence they cite it is difficult to disagree. When once considers the importance of light rail and related transportation methods to 21st century urban development, when cities will only get larger and denser, public transportation must continue to exist. Therefore, the section on how much gentrification actually occurs as a result of light rail development and how much that gentrification actually connects to neighborhood transition and gentrification should be of special interest to readers. The overall conclusion may be a bit dissatisfying, but is by no means surprising. There simply isn’t any consensus as to whether displacement will occur and what is to blame for it. But, there are indeed health consequences for both those who must leave gentrifying areas and those who are able to hold the course. For researchers of chronic health concerns like hypertension, depression, anxiety, smoking and obesity, the citations offered by the authors will help produce an understanding of health outcomes that takes geographic place seriously.
However, given that some published studies find displacement to be a serious issue and others do not, a bigger question remains, one that the authors or future researchers might consider addressing. The authors acknowledge very early on that transportation policy is a crucial element of preparing cities for the 21st century and addressing problems like inequality, sprawl, pollution and traffic congestion. One is left to wonder: if deindustrialization is bad, and white flight is bad, and vacancy is bad, and suburbanization is bad but reinvestment, gentrification and neighborhood transition is also bad – doesn’t that leave us paralyzed?
Author Response
Review #1,
Thank you for your review of this manuscript. Please see our attached response.
Best,
Jennifer Roberts

Reviewer 2 Report
Dear authors of “The Color of Health: Residential Segregation, Light Rail Transit Developments and Gentrification in the United States”, I appreciate your paper. It is an interesting and important topic especially 4th section which described the impact of gentrification on health, and the manuscript is clear and well written. But I felt some minor changes are required.
First, throughout the paper transit oriented developed are used as a synonym to transit station areas. I felt it needs to be modified, as all the transit station are not TODs. Most of the station areas are just transit adjacent developments. For more details please refer: Renne, J. L., & Ewing, R. (2013). Transit-Oriented Development: An Examination of America’s Transit Precincts in 2000 & 2010. I appreciate if you can distinguish the stations area and TODs. I believe census tracts with improved transit accessibility has high probability of gentrification than the census tracts away from transit and if the census tracts have also had TOD policies in place, it further increases the probability of gentrification.
The section wise detailed comments are as follows:
2.1: Suburbanization and Neighborhood Design: The section is mostly about suburbanization and laws which encouraged racial segregation. Not much about design. I think titles of the section needs to be changed to match the content. Sorry, not able to relate title with the content.
2.3. Urban Planning and Residential Zoning
It would be clearer, if you can define exclusionary zoning ordinance and how it is responsible for current American neighborhood social upgradation briefly. I did not understand quite well, how the transit deserts are the function of TODS. I really appreciate if you can explain it a bit more.
3.2. Transit-Oriented Development and Light Rail Transit
The title needs to be changed. Mostly talking about land premiums around station area. So, I think It would be appropriate to rename it as LRT and real estate market trends or land value uplift something.
3.3. Defining Gentrification and Transit-Induced Gentrification
please refer Davidson, M., & Lees, L. (2005). New-build gentrification and London’s riverside renaissance. Environment and Planning A, 37, 1165–1190. https://doi.org/10.1068/a3739. The definition of gentrification has extended it is something like influx of capital/gentries in inhabited or brownfield located anywhere leading to the direct or exclusionary displacement of existing residents, and socioeconomic uplift of the area. You are focusing only on influx of gentries, in the subsequent waves there are evidence that middle class also replaced by affluent. So, I advise you to use extended definition rather than the initial Glass’s definition which also focuses on exclusionary displacement. “Impoverished neighborhoods and communities of color often bear the brunt of unintended 216 transit-oriented development impacts”. I think it should be improved transit accessibility impact rather than TOD, as I said previously all stations are not TODs. “negative impacts of transit-oriented development, specifically the introduction of LRT stations”. I assume you are referring to Pollack study. But, in his study also he clearly mentioned transit rich neighborhoods. Please modify.
Research Examining Transit-Induced Gentrification “Recent studies have utilized changes in household income, house values, and educational levels of the community surrounding a transit center as measures of TIG”. Please refer Padeiro, M., Louro, A., & Da Costa, N. (2019). Transit-oriented development and gentrification: a systematic review. Transport Reviews. and Chava, J., Newman, P., & Tiwari, R. (2019). Gentrification in New Build and Old Build Transit Oriented Developments: The Case of Bangalore. Urban Research and Practice, 12(4). https://doi.org/10.1080/17535069.2018.1437214. There are several other socioeconomic indicators are adopted to examine transit induced gentrification apart from these three. In second paragraph, Baker, D. M., & Lee, B. (2019). How Does Light Rail Transit ( LRT ) Impact Gentrification ? Evidence from Fourteen US Urbanized Areas. Journal of Planning Education and Research, 39(1), 35–49. https://doi.org/10.1177/0739456X17713619. Bekers study exhibited mixed results varying from region to region. Thus, implying that the impacts of LRT stations varying depending on regional contexts and planning efforts. It would give more clarity to the reader if you include the region context also Conclusions: “Ensure that existing residents remain intact”. I think there is a need to ensure to reduce exclusionary displacement by encouraging mixed-income new developments.Please include reference to the best practices you emphasized in the conclusion section. (Refer Reconnecting America; Center for Community Innovation; Non-Profit Housing Association of Northern California. Transit-Oriented for All: The Case for Mixed-Income Transit-Oriented Communities in the Bay Area; Center for Community Innovation: Berkeley, CA, USA, 2007. and Chava J and Newman P (2016) Stakeholder Deliberation on Developing Affordable Housing Strategies: Towards Inclusive and Sustainable Transit Oriented Developments. Sustainability, 8(10).) Looking forwarded to the final version.
Author Response
Review #2,
Thank you for your review of this manuscript. Please see our attached response.
Best,
Jennifer Roberts
